# SRSF7 and SRSF3 depend on RNA sequencing motifs and secondary structures to regulate Microprocessor

Minh Ngoc Le*, Trung Duc Nguyen*, Tuan Anh Nguyen

**Human Microprocessor cleaves pri-miRNAs to initiate miRNA biogenesis. The accuracy and efficiency of Microprocessor cleavage ensure appropriate miRNA sequence and expression and thus its proper gene regulation. However, Microprocessor cleaves many pri-miRNAs incorrectly, so it requires assistance from many cofactors. For example, SRSF3 enhances Microprocessor cleavage by interacting with the CNNC motif in pri-miRNAs. However, whether SRSF3 can function with other motifs and/or requires the motifs in a certain secondary structure is unknown. In addition, the function of SRSF7 (a paralog of SRSF3) in miRNA biogenesis still needs to be discovered. Here, we demonstrated that SRSF7 could stimulate Microprocessor cleavage. In addition, by conducting high-throughput pri-miRNA cleavage assays for Microprocessor and SRSF7 or SRSF3, we demonstrated that SRSF7 and SRSF3 function with the CRC and CNNC motifs, adopting certain secondary structures. In addition, SRSF7 and SRSF3 affect the Microprocessor cleavage sites in human cells. Our findings demonstrate the roles of SRSF7 in miRNA biogenesis and provide a comprehensive view of the molecular mechanism of SRSF7 and SRSF3 in enhancing Microprocessor cleavage.**

## Introduction

MicroRNAs (miRNAs) play an essential role in gene regulation by silencing gene expression (Friedman et al, 2009; Ameres & Zamore, 2013; Jonas & Izaurralde, 2015; Bartel, 2018; Gebert & MacRae, 2019; Kilikevicius et al, 2022). In humans, Microprocessor (MP) is a protein complex consisting of DROSHA and DGCR8, which is responsible for the production of most miRNAs. This complex cleaves primary miRNAs (pri-miRNAs) to generate precursor miRNAs (pre-miRNAs), containing a 5'-end and often a 2-nucleotide (nt) overhang 3'-end. Subsequently, DICER binds to the ends of the pre-miRNAs and cleaves them at 21–22 nt from the ends to produce miRNA duplexes. Finally, Argonaute protein selects one strand of each miRNA duplex to be the mature miRNA. This miRNA guides Argonaute to target mRNAs, inhibiting translation or stimulating mRNA degradation (Ha & Kim, 2014; Bartel, 2018; Nguyen et al, 2019). As the accuracy of MP cleavage determines the ends of the pre-miRNAs, it plays an important role in determining the sequences of the resulting miRNAs and their gene-silencing functions.

The DROSHA, a component of human MP, is responsible for determining the cleavage sites of the complex. It recognizes the ssRNA/dsRNA (single-stranded RNA/double-stranded RNA) junction in pri-miRNAs and (usually) cleaves them at ~13 bp from this junction (Nguyen et al, 2015). In *Caenorhabditis elegans*, DROSHA often cleaves pri-miRNAs at ~16 bp from the junction (Nguyen et al, 2023). Because pri-miRNAs contain two ssRNA/dsRNA junctions at the basal and apical sides, DROSHA can cleave them at either junction (Fig 1A). Basal cleavage results in the production of miRNAs, so this is called productive cleavage, whereas apical cleavage disrupts the miRNA sequence, so this is called unproductive cleavage (Nguyen et al, 2015). However, DROSHA does not always measure 13 bp exactly, but it might cleave pri-miRNAs at multiple positions at the basal junction; these are called alternative (or basal alternative) cleavages (Kim et al, 2021). The alternative cleavages of DROSHA generate different pre-miRNA sequences with different 5'- and 3'-ends, thereby producing multiple miRNA sequences from one pri-miRNA. A change at the 5'-end of pre-miRNA shifts the seed sequence of the 5p miRNA and thus might cause a significant impact on the silencing activity of miRNA (Bofill-De Ros et al, 2019; Park et al, 2022). On the contrary, an alteration at the 3'-end of a pre-miRNA might change the seed sequence of 3p miRNA depending on how it affects the DICER cleavage site. Because of this multi-cleavage feature of DROSHA, many mechanisms are needed to ensure that this enzyme cleaves pri-miRNAs precisely to generate an exact miRNA sequence.

The first group of mechanisms ensures that DROSHA selects the basal junction rather than the apical junction (Auyeung et al, 2013; Ha & Kim, 2014; Fang & Bartel, 2015; Nguyen et al, 2015, 2018, 2019; Bartel, 2018; Kim et al, 2018, 2021; Kwon et al, 2019; Dang et al, 2020; Le et al, 2020; Li et al, 2020, 2021). This group can be classified into two types. Type I facilitates the interaction of DROSHA with the basal junction, and type II prevents DROSHA from binding to the apical junction. The type I mechanism involves UG and mGHG motifs in

Division of Life Science, The Hong Kong University of Science & Technology, Hong Kong, China

Correspondence: tuananh@ust.hk
*Minh Ngoc Le and Trung Duc Nguyen contributed equally to this work

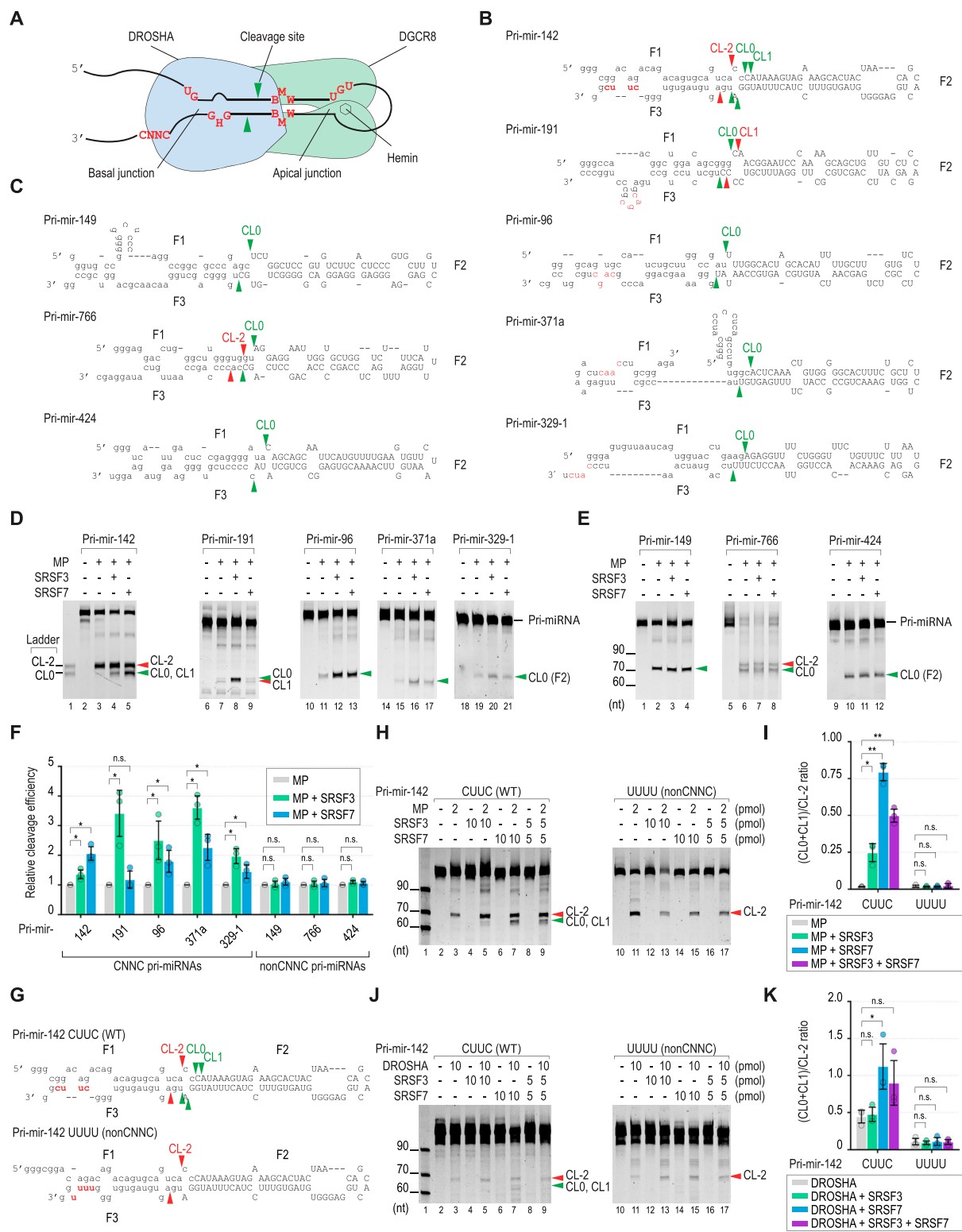

**Figure 1. SRSF7 stimulates DROSHA to cleave CNNC-pri-miRNAs.**

**(A)** Pri-miRNA structure. The green arrowheads indicate the DROSHA cleavage sites. The letters in red are the motifs of pri-miRNAs that are important for the MP cleavage. **(B)** Sequences and structures of tested CNNC-pri-miRNAs. The CNNC motifs are in red. The green and red arrowheads indicate canonical basal cleavages and alternative basal cleavages, respectively. The capital letters and F2 represent pre-miRNA. **(C)** Sequences and structures of tested non-CNNC-pri-miRNAs. The green and red arrowheads indicate canonical basal cleavages and alternative basal cleavages, respectively. The capital letters and F2 represent pre-miRNA. **(D, E)** Pri-miRNA cleavage assays. 2.5 pmol of each pri-miRNA was incubated with 2.5 pmol MP (comprising the NLSD3-DGCR8 complex) alone or with 10 pmol SRSF7 or SRSF3. The pre-miR-142_CL-2

pri-miRNAs that directly interact with DROSHA to facilitate its interaction with the basal junction (Fig 1A) (Auyeung et al, 2013; Fang & Bartel, 2015; Nguyen et al, 2015; Kwon et al, 2019). In addition, SRSF3 binds to the CNNC motif, which results in DROSHA being recruited to the basal junction (Fig 1A) (Auyeung et al, 2013; Kim et al, 2018, 2021). The CNNC motif (N is any nt) is located in the 3′-segment of pri-miRNAs and ~16–18 nt from the 3′-cleavage site of DROSHA. In humans, 59.4% of pri-miRNAs contain the CNNC motif, and the CNNC motif is conserved in many animal pri-miRNAs (Auyeung et al, 2013). The type II mechanism involves DGCR8 binding to the apical loop and, in this way, blocking apical cleavage by DROSHA (Nguyen et al, 2015, 2018; Dang et al, 2020). In addition, the UGU motif in pri-miRNAs and a small molecule, hemin, are known to strengthen the DGCR8–loop interaction, further blocking DROSHA apical cleavage (Fig 1A) (Quick-Cleveland et al, 2014; Nguyen et al, 2015, 2018; Partin et al, 2017; Dang et al, 2020). Furthermore, the midBMW10–12 motif (bulges, mismatches, and wobble base pairs located in the middle of the pri-miRNA upper stem) also inhibits apical cleavage by DROSHA (Fig 1A) (Li et al, 2020, 2021). However, there are several mechanisms, which negatively affect the productive cleavage of DROSHA. For example, seedBMW (bulges, mismatches, and wobble base pairs located in the seed sequence of miRNAs) (Li et al, 2020, 2021) inhibits the productive cleavage of DROSHA. In addition, the mislocalization of mGHG (amGHG) recruits DROSHA to the apical junction (Le et al, 2020), and internal loops in the lower stem increase the single cleavage by DROSHA and thus reduce its productive cleavage mechanism (Nguyen et al, 2020, 2021).

The second group of mechanisms guarantees that DROSHA selects the correct cleavage positions once bound to the basal junction. BMW (bulges, mismatches, and wobble base pairs) located at positions 7–9 from the cleavage sites (called midBMW7–9) is reported to induce DROSHA to cleave at 12 bp from the basal junction in several pri-miRNAs (Li et al, 2020, 2021). The position of mGHG also influences the location of DROSHA cleavage such that it cleaves at 12 bp or 14 bp but not at 13 bp from the basal junction (Fang & Bartel, 2015; Kwon et al, 2019). In addition, SRSF3 also affects the site of DROSHA cleavage in many pri-miRNAs (Kim et al, 2018, 2021). For example, in pri-miR-142, SRSF3 induces DROSHA to cleave at two different sites at the basal junction.

SRSF1–SRSF12 constitute a highly conserved serine/arginine-rich (S/R-rich) splicing factor family. Various homologs of these factors were found in metazoans (Barbosa-Morais et al, 2006). Each member contains one or two RNA recognition motifs (RRMs) at the N-terminal region and an S/R-rich domain (RS domain) at the C-terminal region (Zahler et al, 1993; Long & Caceres, 2009; Busch & Hertel, 2012). Although the RRMs of the SRSF family are highly conserved, their RS domains are more diverse in both sequence and length. Several SRSFs are known to be involved in miRNA biogenesis. For example, the overexpression of SRSF1 in HeLa cells changes the expression of many miRNAs, including miR-221, miR-222, miR-7, and miR-29b (Wu et al, 2010). Furthermore, SRSF1 is suggested to regulate the MP cleavage of pri-miR-7 by binding to its lower stem (Wu et al, 2010). SRSF1 also negatively regulates the expression of miR-222 by enhancing the inclusion of a 44-bp mini exon of MIR222HG (Sun et al, 2020). In addition, both SRSF1 and SRSF2 affect the expression of non-canonical miRNAs. The overexpression of SRSF1 and SRSF2 in HCT116 cells enhances the expression of miR-1229-3p and miR-1227-3p, respectively, although the mechanism is unknown (Butkytè et al, 2016). Furthermore, SRSF3 is known to bind to the CNNC motif in pri-miRNAs and recruits DROSHA to the basal junction. Therefore, SRSF3 affects both the cleavage efficiency and accuracy of MP (Auyeung et al, 2013; Fernandez et al, 2017; Kim et al, 2018, 2021).

SRSF1 and SRSF3 have been demonstrated to regulate miRNA biogenesis by controlling the cleavage of pri-miRNAs by MP. However, it is not known whether any other SRSF proteins also function in miRNA biogenesis by governing pri-miRNA cleavage. Here, we investigated the ability of seven SRSF proteins to regulate pri-miRNA cleavage by MP and revealed that SRSF7 could stimulate cleavage. We conducted high-throughput pri-miRNA cleavage assays and demonstrated that SRSF7 and SRSF3 function with CRC (R is A or G) and CNNC motifs in certain structures to enhance MP cleavage by strengthening the interaction between DROSHA and the basal junction of pri-miRNAs. In addition, using pri-miR-142 as a substrate model, we showed that SRSF7 and SRSF3 could control DROSHA cleavage sites in human cells. Together, our findings establish a function for SRSF7 in miRNA biogenesis and provide a more comprehensive view of the mechanisms of SRSF7 and SRSF3 in stimulating the MP cleavage.

# Results

### SRSF7 stimulates DROSHA to cleave CNNC-pri-miRNAs

We successfully cloned and purified seven of the 12 SRSF proteins, and these were shown to contain different structures such that SRSF2, SRSF3, and SRSF8 contain one RRM domain; SRSF1, SRSF5, and SRSF9 contain RRM and RRMH (RRM homology); and SRSF7 contains one RRM and a Zn finger (Fig S1A–C). We also purified MP containing the D3 fragment (amino acids 390–1,365) of DROSHA and

and pre-miR-142_CL0 were synthesized as described in the Materials and Methods section and used as the RNA ladders in lane 1. F2 fragments were cleaved products. **(F)** Relative pri-miRNA cleavage efficiency of MP was calculated from three repeated experiments, as shown in panels (D, E) and in Fig S1G and H. One-tailed, two-sample (assuming unequal variances) $t$ tests were used to calculate the $P$-value. *$P \le 0.05$; **$P \le 0.005$; ***$P \le 0.0005$; and n.s., not significant. The exact $P$-values are presented in Table S6. **(G)** Sequences and structures of pri-miR-142. The CNNC motif and its mutated nt are shown in red. The capital letters and F2 represent pre-miRNA. **(H)** Pri-miRNA cleavage assays. 2.5 pmol of each pri-miRNA was incubated with 2 pmol MP (comprising the NLSD3-DGCR8 complex) alone or with the indicated amounts of SRSF7 and/or SRSF3. **(I)** Ratios of canonical to alternative cleavages of MP in pri-miR-142 were calculated from three repeated experiments, as shown in panel (H) and Fig S1I. One-tailed, two-sample (assuming unequal variances) $t$ tests were used to calculate the $P$-value. *$P \le 0.05$; **$P \le 0.005$; ***$P \le 0.0005$; and n.s., not significant. The exact $P$-values are presented in Table S6. **(J)** Pri-miRNA cleavage assays. 2.5 pmol pri-miRNA was incubated with 10 pmol DROSHA (comprising the D3-G2 complex) alone or with indicated amounts of SRSF7 and/or SRSF3. **(K)** Ratios of canonical to alternative cleavages of DROSHA in pri-miR-142 were calculated from three repeated experiments, as shown in panel (J) and Fig S1J. One-tailed, two-sample (assuming unequal variances) $t$ tests were used to calculate the $P$-value. *$P \le 0.05$; **$P \le 0.005$; ***$P \le 0.0005$; and n.s., not significant. The exact $P$-values are presented in Table S6.

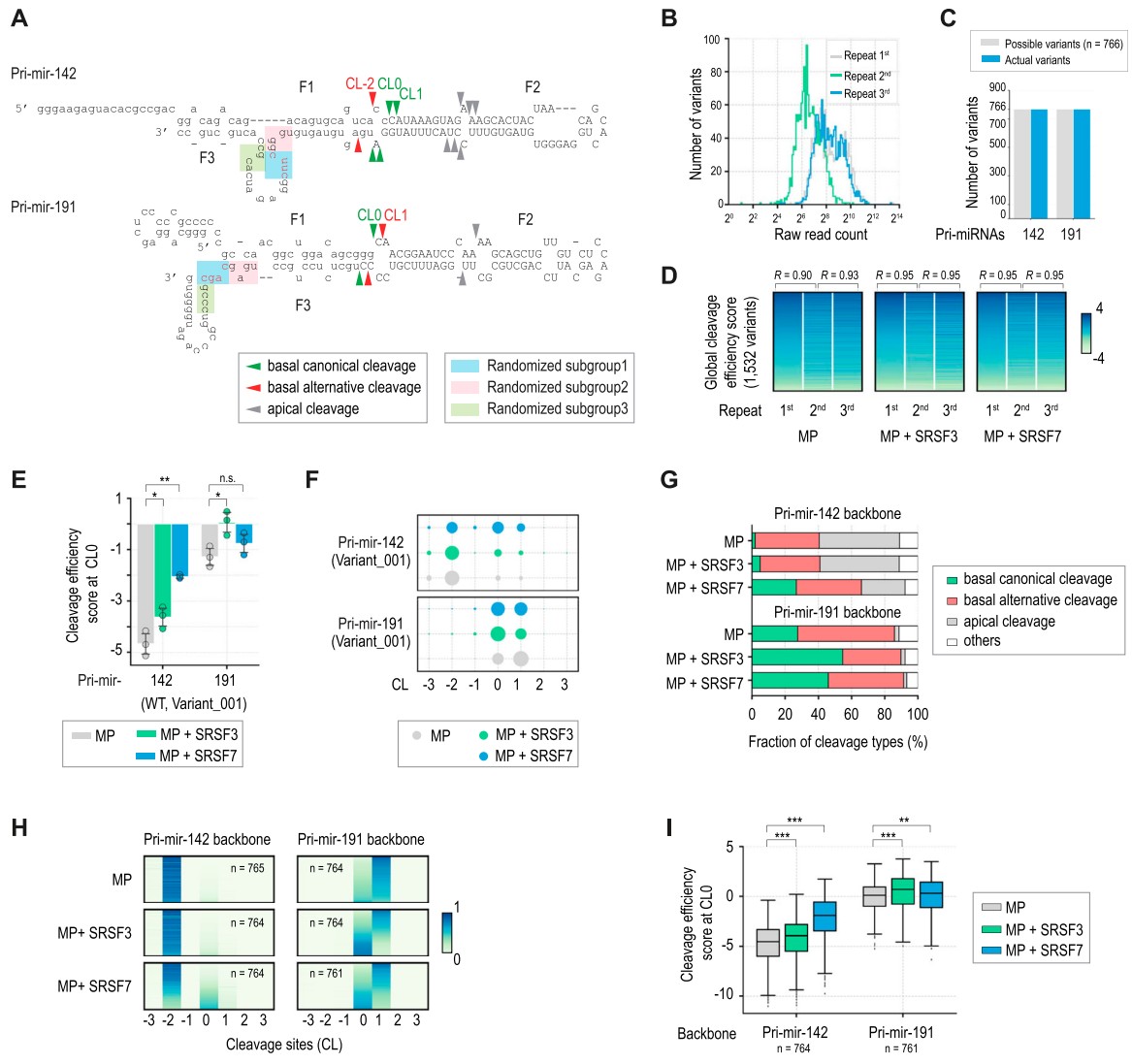

**Figure 2. High-throughput pri-mRNA cleavage assay (HT-PriCA).**
**(A)** Sequences and structures of pri-miR-142 and pri-miR-191. The green, red, and gray arrowheads indicate the basal canonical, basal alternative, and apical cleavage sites of MP, respectively. The blue, pink, and green rectangles highlight randomized nt. Each group contains four randomized nt. The CNNC motif is shown in red. The capital letters represent the pre-miRNA sequences. **(B)** Distribution of raw read counts in the original substrate samples from three repeated HT-PriCA. **(C)** Pri-miRNA variant numbers in the original substrate samples were obtained by NGS. **(D)** Reproducibility of HT-PriCA was determined by its global cleavage efficiency score. R is Pearson's correlation coefficient. **(E)** Cleavage efficiency scores at CL0 of WT pri-miRNA variants (Variant_001) found by HT-PriCA. One-tailed, two-sample (assuming unequal variances) $t$ tests were used to calculate the $P$-value. *$P \leq 0.05$; **$P \leq 0.005$; ***$P \leq 0.0005$; and n.s., not significant. **(F)** Cleavage accuracy scores of the WT pri-miRNA variants (Variant_001) found in HT-PriCA. The size of the circles indicates the relative accuracy scores. **(G)** Read fraction of the different cleavage types (i.e., basal canonical, basal alternative, and apical cleavages), detected by HT-PriCA. The basal and apical cleavage sites ranged from CL-3 to CL3, and from CL8 to CL14, respectively. "Others" indicates the cleavage sites beyond these two ranges (e.g., CL-4 and CL5). Basal canonical cleavage sites are CL0 and CL1 for pri-miR-142 and CL0 for pri-miR-191, as shown in panel (A). **(H)** Cleavage accuracy scores at different cleavage sites for each pri-miRNA variant found in HT-PriCA. The cleavage accuracy scores are represented according to the color scale shown on the right of the figure. Each line represents one pri-miRNA variant. **(I)** Cleavage efficiency scores of CL0 in the pri-miRNA variants found in HT-PriCA. The $P$-value was calculated by one-tailed Wilcoxon rank-sum tests. *$P \leq 0.05$; **$P \leq 0.005$; ***$P \leq 0.0005$; and n.s., not significant.

full-length DGCR8 (Fig S1B and C). We selected pri-miR-142 as a substrate model to examine the effects of these SRSF proteins on MP cleavage because SRSF3 is known to interact with DROSHA to induce alternative cleavages in this pri-miRNA (Kim et al, 2018, 2021). Here, we named the pre-miRNA isoforms (F2, MP cleavage products) by CLx, where "CL" stands for "cleavage," and "x" is the cleavage site position. CL0 is a canonical cleavage site of MP in each pri-miRNA, annotated in MirGeneDB (Fromm et al, 2020) (Fig 1B and C). Interestingly, the assay revealed that among the seven SRSFs, SRSF7

and SRSF3 enhanced MP to cleave at CL0 and CL1 (Fig S1D). The other SRSF proteins seemed to stimulate MP to cleave at CL-2 and CL0 slightly (Fig S1D).

As SRSF3 is known to stimulate MP to cleave CNNC-pri-miRNAs (Auyeung et al, 2013; Fernandez et al, 2017; Kim et al, 2018, 2021), and SRSF7 has been shown to bind to the CNNC motif in pri-miRNAs (Auyeung et al, 2013), we hypothesized that SRSF7 also stimulates MP activity by interacting with this motif. Thus, we further examined the effects of SRSF7 in four CNNC-pri-miRNAs (Fig 1B) and three

non-CNNC-pri-miRNAs (Fig 1C). We chose 10 pmol of SRSF7 in these assays because SRSF7 optimally stimulated MP in a range of 8–16 pmol (Fig S1E and F). We demonstrated that SRSF7 and SRSF3 stimulated the MP cleavage in CNNC-pri-miRNAs, except that there was no effect of SRSF7 in pri-miR-191 (Figs 1D and F and S1G). In contrast, SRSF7 and SRSF3 did not enhance the activity of MP in non-CNNC-pri-miRNAs (Figs 1E and F and S1H). In addition, SRSF7 and SRSF3 lost their stimulatory effect on the MP cleavage in non-CNNC-pri-miR-142 (Figs 1G–I and S1I). These results indicate that SRSF7 stimulates the MP cleavage in a CNNC-dependent manner. In addition, the stimulatory effect of 5 pmol SRSF7 and 5 pmol SRSF3 on MP was an average of 10 pmol SRSF7 or 10 pmol SRSF3 alone, suggesting that these two proteins likely stimulate MP independently (Figs 1G–I and S1I).

We also showed that SRSF7 changed the cleavage sites of DROSHA in CNNC-pri-miR-142 but not in non-CNNC-pri-miR-142 (Figs 1J and K and S1J). In addition, DROSHA, SRSF7, and pri-miR-142 formed a ternary complex in the electrophoretic mobility shift assays (EMSA), suggesting that both DROSHA and SRSF7 were bound to one RNA molecule (Fig S1K). The density of the DROSHA/SRSF7/pri-miR-142–supershifted band was higher than that of DROSHA/RNA (Fig S1K, comparing lanes 4 and 5 with lane 1), suggesting that SRSF7 strengthens the interaction between DROSHA and pri-miRNA. Furthermore, SRSF7 reduced the unproductive cleavage of DROSHA, which occurred at the apical junction (Fig S1L and M). These results suggest that SRSF7 stimulates the MP cleavage by interacting with the CNNC motif to enhance the interaction between DROSHA and the basal junction of CNNC-containing pri-miRNA.

### High-throughput pri-mRNA cleavage assays for MP ± SRSF7 or SRSF3

To understand how the RNA-interacting features of SRSF7 and SRSF3 function to stimulate pri-miRNA cleavage in further detail, we conducted a series of high-throughput (HT) pri-miRNA cleavage assays (HT-priCA). In these assays, we cleaved two pri-miRNAs containing randomized nt in CNNC and its neighboring regions, either with MP alone or with MP plus SRSF7 or SRSF3. We produced three randomized groups for each pri-miRNA (Fig 2A) and expected to generate 766 (3 × $4^4$−2) variants (the WT variant appeared three times in the three groups) for each pri-miRNA, and hence 1,532 variants for two pri-miRNAs. After HT-priCA, we cloned the original substrates (OS) and the cleaved products (F3 fragments), both of which contained randomized sequences, as described in the Materials and Methods section (Fig S2A). The resulting cloned DNA libraries were sequenced using next-generation sequencing (NGS). We obtained more than 100 reads for each variant in at least one of the three repeats and got all the expected 766 variants for each pri-miRNA backbone (Fig 2B and C). An estimation of the global cleavage efficiency for each variant (see the Materials and Methods section for details) confirmed the reproducibility between any two of the three repeats (Fig 2D).

To validate our HT-priCA, we calculated the cleavage efficiency and accuracy of MP alone or MP plus SRSF7 or SRSF3 for the WT variant of each pri-miRNA (Figs 2E and F and S2B) and found that SRSF7 strongly induced the CL0 and CL1 cleavages in pri-miR-142, and SRSF3 effectively stimulated the CL0 cleavage in pri-miR-191. These results are consistent with those we obtained in the pri-

miRNA cleavage assays conducted for the individual pri-miRNAs, as shown in Fig 1D and F. This indicates that the HT-priCA, RNA cloning, and sequencing were conducted correctly to observe the effects of SRSF7 and SRSF3 on the MP cleavage.

Next, we found that MP cleaved pri-miR-191 variants mostly at the basal junction, whereas it cleaved pri-miR-142 variants equally at the apical and basal junctions (Fig 2G). SRSF7 increased the basal cleavage of MP, especially for the pri-miR-142 variants, suggesting that this cofactor helps to strengthen the interaction between DROSHA and the basal junction of pri-miRNAs (Fig 2G). We also showed that MP cleaved the pri-miRNA variants at multiple sites at the basal junction (resulting in basal canonical cleavage and basal alternative cleavage) and that SRSF7 and SRSF3 affected the cleavage sites of MP differently (Fig 2G and H). For example, SRSF3 increased the accuracy and efficiency of CL0 cleavage in pri-miR-191 more than SRSF7 (Fig 2G–I). However, SRSF7 enhanced the accuracy and efficiency of the CL0 and CL1 cleavages in pri-miR-142 variants far more than SRSF3 (Fig 2G–I). Pri-miR-142 is reported to produce three miRNA isoforms from CL0, CL1, and CL-2 (Chen et al, 2004; Wu et al, 2009). However, MP alone mainly cleaves this pri-miRNA at CL-2 (Kim et al, 2018, 2021). Thus, the effect of SRSF7 observed in our study together with that of SRSF3 reported previously (Kim et al, 2021) might explain how pri-miR-142 can produce three miRNA isoforms in human cells.

### SRSF7 and SRSF3 enhance MP cleavage in pri-miRNAs containing CNNC and 17-CRC motifs

Next, we analyzed 256 variants of each pri-miRNA that were randomized in the CNNC position. We found that in two backbones, both SRSF7 and SRSF3 stimulated MP to cleave CNNC-pri-miRNA variants with higher efficiency and accuracy than non-CNNC-pri-miRNA variants, suggesting that both of these cofactors require the CNNC motif for their stimulatory effect (Fig 3A–H). Interestingly, our HT-priCA also identified many non-CNNC motifs, such as DCRC and CRCD (D being A, U, or G, and R being A or G), which could also facilitate the stimulatory effect of SRSF7 and SRSF3 in two pri-miRNA backbones (Fig 3A–H). The CRC motif in both DCRC and CRCD is located at the −17 position from the cleavage site of DROSHA on the 3′-strand, so-called 17-CRC. Consistently, we found that this 17-CRC motif enhanced the cleavage accuracy of MP from human pri-miRNA cleavage library data (Kim et al, 2021) (Fig S3A–B). These non-CNNC motifs even enhanced the cleavage activity of MP more than several CNNC motifs, including CUCC, CGGC, and CCCC (Fig 3A–D).

We selected two CNNC-pri-miRNAs, pri-miR-142 (used in the HT cleavage assays) and pri-miR-411 (not used in the HT cleavage assays), which showed distinctive cleaved products by MP in the urea–PAGE, to verify the effect of 17-CRC. We confirmed that SRSF7 and SRSF3 enhanced the cleavage accuracy of MP in the CNNC and 17-CRC variants, but not in the non-CNNC variants (Figs 3I–N and S3C and D). Furthermore, SRSF7 and SRSF3 had a weaker stimulatory effect with the 17-CRC motif than with the CNNC motif (Fig 3K and N). These findings are consistent with our HT-priCA data (Fig 3E–H).

### The RNA secondary structures required for the function of SRSF7 and SRSF3

We next analyzed the secondary structures of the CNNC-pri-miRNA variants and classified them into eight different structures

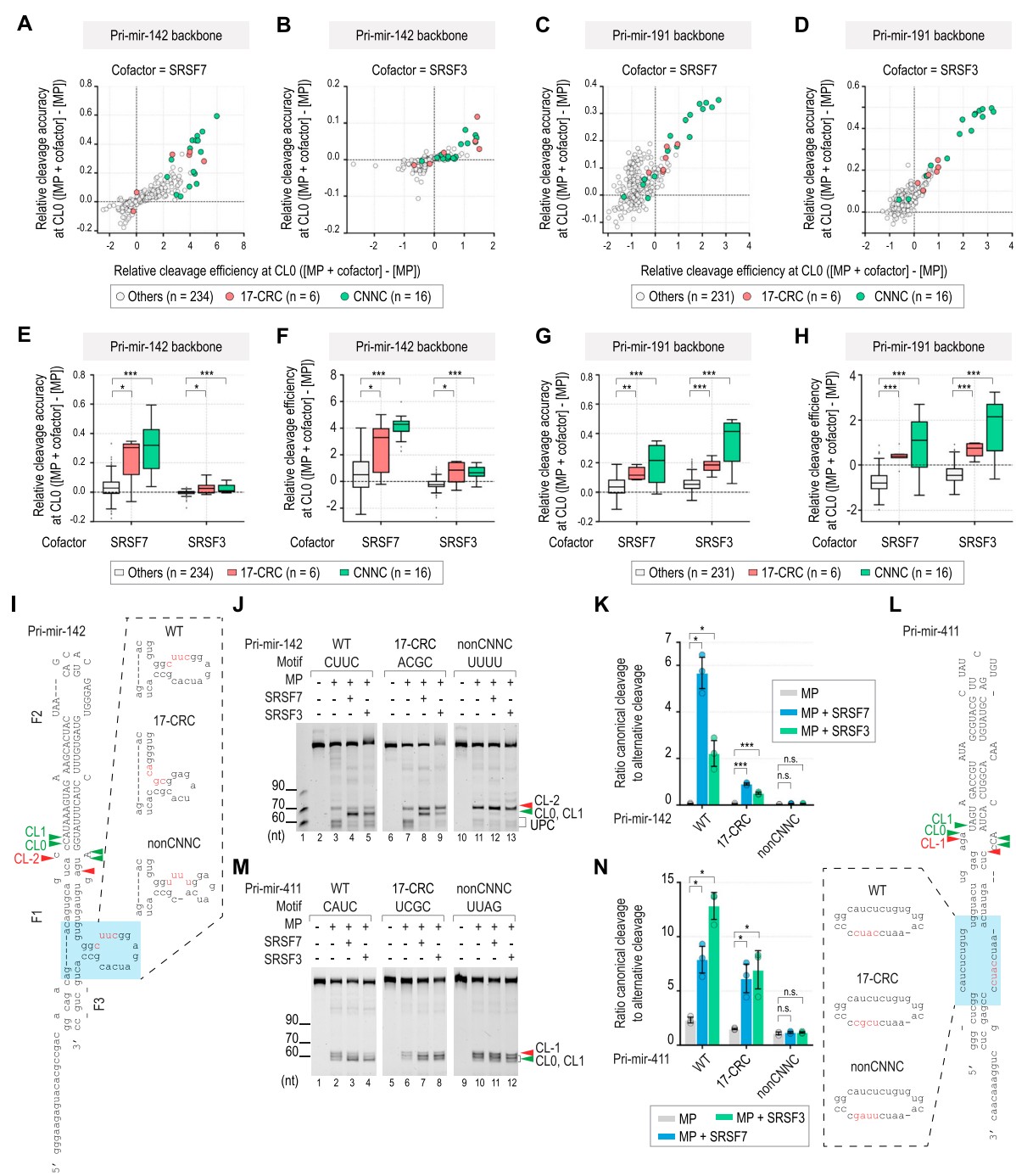

**Figure 3. SRSF7 and SRSF3 enhance MP cleavage in pri-miRNAs containing CNNC and 17-CRC motifs.**
**(A, B, C, D)** Relative cleavage efficiency (x-axis) and accuracy (y-axis) at CL0 were calculated for each pri-miR-142 (in A and B) or pri-miR-191 (in C and D) variant. The green and red circles indicate pri-miRNA variants that contain the CNNC and CRC motifs, respectively. The gray circles indicate pri-miRNA variants that contain motifs other than CNNC or CRC. R is A or G. **(E, F, G, H)** Relative cleavage accuracy and efficiency at CL0 were calculated for each of three pri-miRNA groups, which contained CNNC, 17-CRC, or others. R is A or G. The *P*-value was calculated by one-tailed Wilcoxon rank-sum tests. *$P \leq 0.05$; **$P \leq 0.005$; ***$P \leq 0.0005$; and n.s., not significant. **(I, L)** Sequences and structures of pri-miR-142 (I) and pri-miR-411 (L). The CNNC motif and its mutated nt are in red. The green and red arrowheads indicate basal canonical cleavages and basal alternative cleavages, respectively. The capital letters represent pre-miRNA. **(J, M)** Pri-miRNA cleavage assays. (J) 2 pmol pri-miR-142 were incubated with 2.5 pmol MP alone or with 20 pmol SRSF7 or SRSF3. (M) 2 pmol pri-miR-411 were incubated with 1.25 pmol MP alone or with 2.5 pmol SRSF7 or SRSF3. The green and red arrowheads indicate basal canonical cleavages and basal alternative cleavages, respectively. UPC, unproductive cleavages. **(K, N)** Graphs showing the canonical-to-alternative cleavage ratios of MP in (K) pri-miR-142 or (N) pri-miR-411. These were calculated from the three repeated experiments as shown in panel (J) (and Fig S3C) and panel (M) (and Fig S3D), respectively. The RNA band intensities were estimated using Image Lab v6.0.1. Two-tailed, two-sample (assuming unequal variances) *t* tests were used to calculate the *P*-value. *$P \leq 0.05$; **$P \leq 0.005$; ***$P \leq 0.0005$; and n.s., not significant. The exact *P*-values are presented in Table S7.

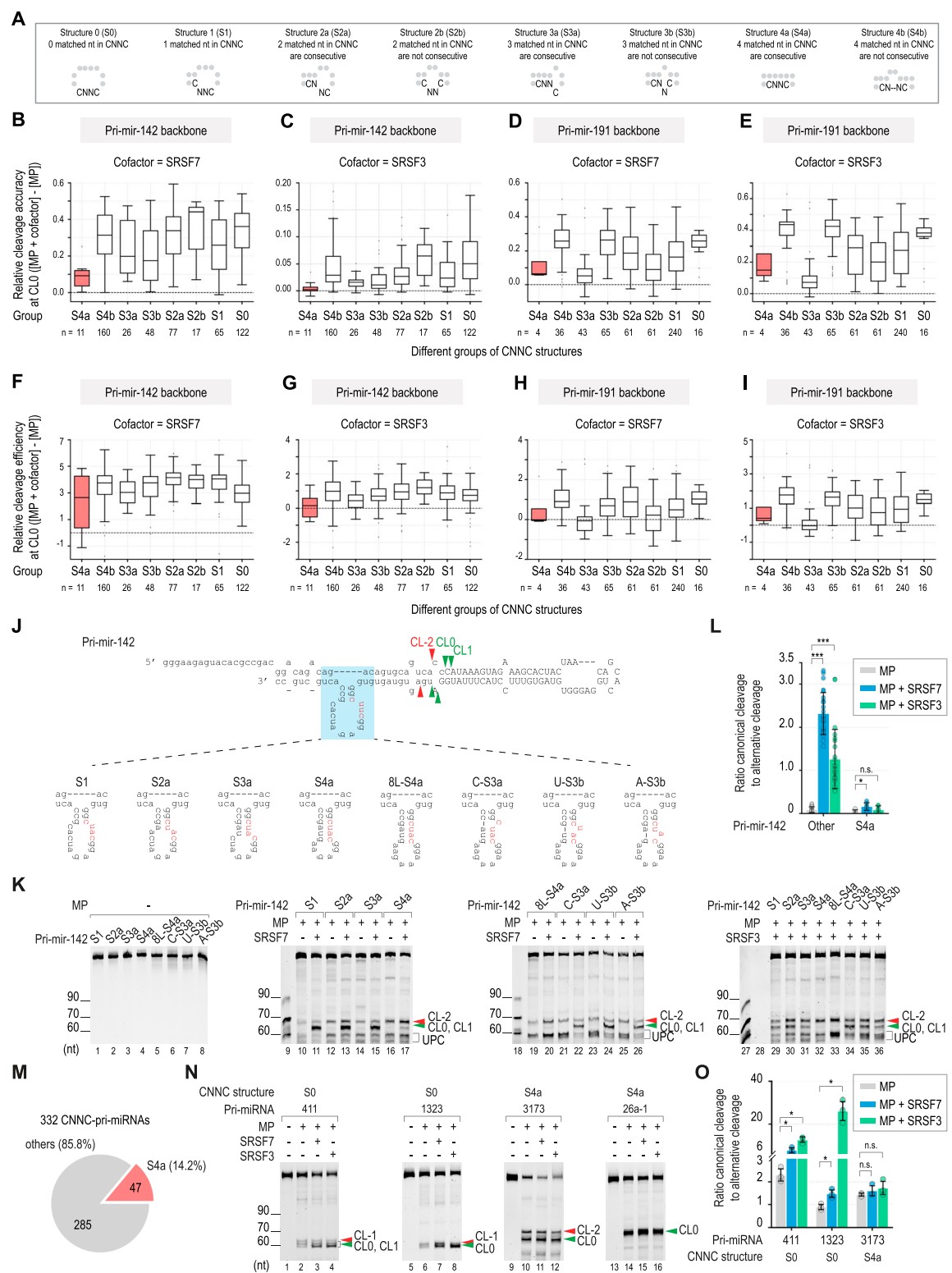

**Figure 4. RNA secondary structures required for the function of SRSF7 and SRSF3.**
**(A)** Secondary structures of CNNC. These were classified as S0, S1, S2, S3, and S4, containing 0–4 nt base pairs, respectively. The S2, S3, and S4 structures were further divided into two subgroups, S2a and S2b, S3a and S3b, and S4a and S4b, such that the "a" and "b" indices indicate that the base-paired nt in CNNC was consecutive or not, respectively. **(B, C, D, E, F, G, H, I)** Relative (B, C, D, E) cleavage accuracy and (F, G, H, I) efficiency at CL0 of each CNNC structure with MP alone or with MP plus SRSF7 or SRSF3. **(J)** Schematic to show the pri-miR-142 variants containing different CNNC structures. The CNNC motifs are in red. The green and red arrowheads indicate basal canonical cleavages and basal alternative cleavages, respectively. The capital letters represent pre-miRNA. **(K, N)** Pri-miRNA cleavage assays. (K) 2 pmol pri-miR-142 or (N) 2 pmol

containing different numbers of base pairs in the CNNC motif (Fig 4A). We then calculated the cleavage accuracy and efficiency for the two pri-miRNA backbones containing each of these structures. We found that the stimulatory effect of SRSF7 and SRSF3 was substantially reduced in the S4a structure in both pri-miR-142 and pri-miR-191 (Fig 4B–I). These results suggest that SRSF7 and SRSF3 can interact with CNNC motifs that contain at least one unbase-paired nt but cannot interact with fully base-paired CNNC motifs.

To verify our HT-priCA findings, we synthesized four pri-miR-142 variants containing S1, S2a, S3a, and S4a (Fig 4J). We showed that SRSF7 and SRSF3 induced MP to cleave more at CL0 and CL1 in the three pri-miR-142 structures (S1, S2a, and S3a), which contained at least one unbase-paired nt in the CNNC motif. In contrast, they did not stimulate MP cleavage in pri-miR-142-S4a (Figs 4K and S4A–C). We also included the S0 structure in this examination by adding one more base pair upstream of the CNNC of pri-miR-142 (generating pri-miR-142-plus-S0) to stabilize the short stem–loop structure that contained this motif (Fig S4D). We found that SRSF7 and SRSF3 retain their MP stimulatory effect in both the pri-miR-142-plus-S0 and plus-S1 structures (Fig S4D–F). We conducted a SHAPE-MaP (selective 2′-hydroxyl acylation analyzed by primer extension and mutational profiling) (Siegfried et al, 2014) for pri-miR-142_S1 and pri-miR-142_S4a to experimentally confirm their RNA structures (Fig S4G). As expected, most of the unpaired nt in the loop showed a high SHAPE reactivity. In contrast, most of paired nt in the stem displayed a low SHAPE reactivity. We then used the SHAPE reactivity of these two pri-miR-142 variants to obtain their RNA structures from the RNAstructure web server (Reuter & Mathews, 2010). The obtained structures were similar to the predicted ones by RNAfold. All four nt of the CUAC motif in pri-miR-142_S4a were in base pairs, whereas the last three nt of the CUAC motif in pri-miR-142_S1 were not paired (Fig S4G).

However, we realized that as the number of base-paired nt increased, the loop size near the CNNC region was shortened from S1 to S4a. To exclude the possibility that the small loop size of S4a affected the activity of SRSF7 and SRSF3, we synthesized pri-miR-142-8L-S4a, which contained the same S4a structure and 8-nt loop as pri-miR-142-S3a (Fig 4J). We again showed that pri-miR-142-8L-S4a reduced the activity of SRSF7 and SRSF3 compared with pri-miR-142-S3a (Figs 4K and S4A–C). This indicates that the key factor that reduces the activity of SRSF7 and SRSF3 is the S4a structure rather than the loop size. Furthermore, when comparing pri-miR-142-8L-S4a and pri-miR-142-S3a, because unpairing the fourth C in the CNNC motif significantly increased the stimulatory effect of SRSF7 and SRSF3 (Figs 4J and L and S4C), we further investigated the effect of unpairing the first three nt in the CNNC motif. We interrupted the continuity of four base pairs in pri-miR-142-8L-S4a by making one of the paired nt in CNNC into a bulge, thus generating pri-miR-142-C-S3a, pri-miR-142-U-S3b, and pri-miR-142-A-S3b (Fig 4J). We observed that a single-nt bulge in any nt in the CUAC motif was sufficient to activate the stimulatory effects of SRSF7 and SRSF3 on the MP cleavage (Figs 4J and K and S4A–C). Together,

these results demonstrated that SRSF7 and SRSF3 have a stimulatory effect on the CNNC motif adopting other structures but not S4a (Fig 4L).

Next, we analyzed the structures of human pri-miRNAs and identified ~14.2% CNNC-pri-miRNAs with the S4a structure (Fig 4M). We selected two S0 CNNC-pri-miRNAs (pri-miR-411 and pri-miR-1323) and two S4a CNNC-pri-miRNAs (pri-miR-3173 and pri-miR-26a-1) and examined their cleavage with MP alone or with MP plus SRSF7 or SRSF3 (Fig S4H). As expected, we found that SRSF7 and SRSF3 stimulated the MP cleavage accuracy in S0 CNNC-pri-miRNAs, but did not affect both MP cleavage accuracy and efficiency in S4a CNNC-pri-miRNAs (Figs 4N and O and S4I and J).

### SRSF7 and SRSF3 control the MP cleavage sites in human cells

As SRSF3 is known to change the MP cleavage sites in pri-miR-142 (Kim et al, 2021), we selected this miRNA as a substrate model to examine the cellular effect of SRSF7 on the MP cleavage sites. We used HCT116 DICER-KO cells to exclude the contribution of DICER cleavage when determining the pre-miR-142 isoforms. In addition, pri-miR-142 is normally expressed at very low levels in these cells. Thus, we overexpressed CNNC-pri-miR-142 or non-CNNC-pri-miR-142 in the DICER-KO cells and then sequenced pre-miR-142 from these transfected cells using iLIME-seq (Le et al, 2022). Our iLIME-seq data demonstrated that in the CNNC-pri-miR-142–overexpressed sample, the main pre-miR-142 isoforms were pre-miR-142_CL0 and pre-miR-142_CL1. In contrast, in the non-CNNC-pri-miR-142–overexpressed sample, just one isoform, pre-miR-142_CL-2, was mainly produced (Fig 5A). This indicates that the CNNC motif in pri-miR-142 plays a key role in producing pre-miR-142_CL0 and pre-miR-142_CL1. Our results are also consistent with those from a previous report, where similar assays were conducted in HEK293T cells (Kim et al, 2021).

Next, we coexpressed CNNC-pri-miR-142 with SRSF7 or SRSF3 in the DICER-KO cells. We confirmed the overexpression of the SRSF cofactors by qRT-PCR (Fig 5B) and sequenced pre-miR-142 using iLIME-seq. Our iLIME-seq results showed that SRSF7 and SRSF3 significantly reduced the CL-2/(CL0+CL1) pre-miR-142 ratios (Fig 5C). These changes were consistent with what we found in our in vitro cleavage assays, such that on MP cleavage, SRSF7 and SRSF3 induced the production of CL0 and CL1 relative to CL-2. This indicates that SRSF7 and SRSF3 influence the cleavage sites of MP in human cells.

## Discussion

In the SRSF family, SRSF7 is the most like SRSF3 with regard to its polypeptide sequence, and their RRMs share ~80% amino acid sequence identity (Cavaloc et al, 1994) (Fig S1A). Biochemically,

---

another CNNC-pri-miRNA were incubated with 2.5 pmol MP alone or with 10 pmol SRSF7 or SRSF3. The green and red arrowheads indicate basal canonical cleavages and basal alternative cleavages, respectively. **(L)** Canonical-to-alternative cleavage ratios of MP in pri-miR-142 variants adopting different CNNC structures. These were calculated from the three repeated experiments, as shown in panel (K) and Fig S4A and B. The RNA band intensities were estimated using Image Lab v6.0.1. Two-tailed, two-sample (assuming unequal variances) $t$ tests were used to calculate the $P$-value. *$P ≤ 0.05$; **$P ≤ 0.005$; ***$P ≤ 0.0005$; and n.s., not significant. The exact $P$-values are presented in Table S8. **(M)** Fraction of human CNNC-pri-miRNAs containing CNNC in the S4a structure. **(O)** Canonical-to-alternative cleavage ratios of MP in different CNNC-pri-miRNAs. These were calculated from the three repeated experiments, as shown in panel (N) and Fig S4I. The RNA band intensities were estimated using Image Lab v6.0.1. Two-tailed, two-sample (assuming unequal variances) $t$ tests were used to calculate the $P$-value. *$P ≤ 0.05$; **$P ≤ 0.005$; ***$P ≤ 0.0005$; and n.s., not significant. The exact $P$-values are presented in Table S8.

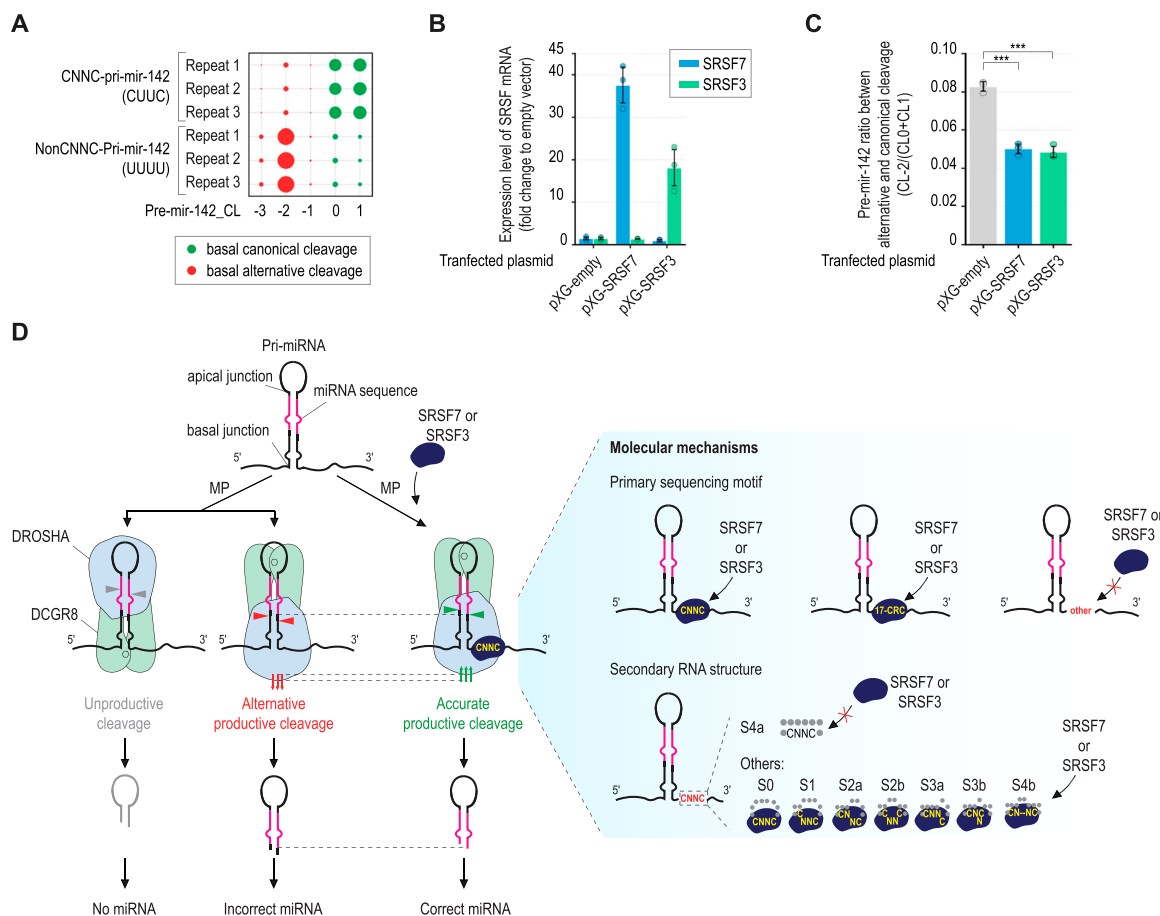

**Figure 5. SRSF7 and SRSF3 control MP cleavage sites in human cells.**
**(A)** Pre-miR-142 isoforms were sequenced from the DICER-KO cells transfected with CNNC-pri-miR-142 or non-CNNC-pri-miR-142 plasmids. **(B)** Expression level of SRSF7 or SRSF3 mRNA in DICER-KO cells, measured by qRT-PCR from three repeated transfection experiments, as shown in panel (A). Two-tailed, two-sample (assuming unequal variances) *t* tests were used to calculate the *P*-value. **(C)** Canonical-to-alternative cleavage ratios of MP in pri-miR-142 in the DICER-KO cells, calculated from n = 3 NGS data. Two-tailed, two-sample (assuming unequal variances) *t* tests were used to calculate the *P*-value. *$P \leq 0.05$; **$P \leq 0.005$; ***$P \leq 0.0005$; and n.s., not significant. The exact *P*-values are presented in Table S9. **(D)** Proposed functions and molecular mechanisms of SRSF7 and SRSF3 in pri-miRNA cleavage by MP. (1) MP is mislocalized at the apical junction of many pri-miRNAs and executes unproductive cleavage. (2) MP binds to the basal junction but mismeasures many pri-miRNAs and so makes alternative cleavages. (3) SRSF7 or SRSF3 interacts with the CNNC and CRC motifs in the 3'-segment of pri-miRNAs; in this way, the interaction between DROSHA and the basal junction is strengthened, which places MP in the correct location for productive cleavage. SRSF7 and SRSF3 also correct the incorrect basal positioning of MP to facilitate its accurate productive cleavage. SRSF7 and SRSF3 can interact with the CNNC motifs in most of the secondary structures, except for S4a, which contains a fully base-paired nt.

SRSF7 and SRSF3 were found to interact with CNNC-containing RNA sequences such that N indicates any nt. Indeed, the RNA-binding motifs of SRSF7 and SRSF3 are AGAC(G/U)ACGA(C/U) and (A/U)C(A/U)(A/U)C, respectively (Hargous et al, 2006). In addition, these two proteins have previously been shown to play critical and redundant roles in mRNA alternative splicing (Zahler et al, 1993; Cavaloc et al, 1994), mRNA export (Huang & Steitz, 2001; Müller-McNicoll et al, 2016), determination of the 3'-UTR of mRNA (Schwich et al, 2021), and the initiation of translation (Bedard et al, 2007; Swartz et al, 2007), as well as the metabolism of other RNAs (Brugiolo et al, 2017). Here, we demonstrated that SRSF7 and SRSF3 also share similar functions in miRNA biogenesis. They can interact with similar motifs (i.e., CNNC and CRC) and use similar structures to strengthen the interaction between DROSHA and the basal junction of pri-miRNAs in vitro. We also showed that in human cells, they affect the MP cleavage sites in pri-miR-142. Both factors have previously been reported to bind to CNNC-pri-miR-

30a in RNA pulldown assays (Auyeung et al, 2013). Together, these data indicate that SRSF7 and SRSF3 have shared roles in miRNA biogenesis (Fig 5D). This study showed the impact of SRSF7 and SRSF3 on the cellular MP cleavage sites in only pri-miR-142; thus, future research should address how these two factors influence the cellular cleavage sites of MP in a genome-wide manner.

Although SRSF7 and SRSF3 mainly function in a similar manner to stimulate the MP activity, in some circumstances, they exhibit distinct activities. For example, SRSF7 has a greater stimulatory effect on MP than SRSF3 in pri-miR-142. In contrast, SRSF3 stimulates MP more than SRSF7 in pri-miR-191, pri-miR-371a, pri-miR-411, and pri-miR-1323. Future study is needed to address these differences from the perspectives of SRSF7/3, MP, or RNAs. For example, these two SRSF proteins might also have different RNA-binding affinities to other parts of pri-miRNAs, including the stem region. Alternatively, these differences might be caused by differential ways MP interacts with these RNAs.

Previously reported SRSF3-RNA structure models, SELEX and iCLIP experiments, and splice array analysis of SRSF3 for human genes suggest that this cofactor interacts with the CNNC motif in ssRNA and the first C is important for SRSF3 recognition (Schaal & Maniatis, 1999; Hargous et al, 2006; Änkö et al, 2012; Ajiro et al, 2016). However, by investigating the RNA sequences in more detail, our data indicate that SRSF7 and SRSF3 work with many CNNC motifs in which a maximum of 3 nt are in base pairs to stimulate MP. This suggests that SRSF7 and SRSF3 might bind to CNNC and locally unwind a partially base-paired CNNC into a single-stranded CNNC as part of their interaction.

Our studies show that SRSF7 and SRSF3 can also interact with pri-miRNAs containing non-CNNC, such as the 17-CRC motif, suggesting that their substrate reservoir might be larger than previously thought. However, they cannot interact with pri-miRNAs that have fully base-paired CNNC motifs. Thus, the secondary structures should be considered when interpreting the CNNC substrates of these proteins.

# Materials and Methods

### Protein expression and purification

The coding sequences for SRSF1, SRSF2, SRSF3, SRSF5, SRSF7, SRSF8, and SRSF9 proteins in humans were amplified by PCR using cDNA obtained from total RNAs of human HCT116 cells. The PCR-amplified DNAs were inserted into human expression plasmid, pXG, used in previous studies (Nguyen et al, 2015). As a result, SRSF proteins were fused with GFP and 10×His-tag at their C-termini. The information of the primers used in the cloning experiments is presented in Table S1.

The seven SRSF proteins were expressed and purified in a similar way. pXG-SRSF plasmids were transfected into 50 of 100-mm dishes of HEK293E cells. The transfected cells were grown for 3 d and then collected by centrifuge. The collected cell pellets were resuspended in T500 buffer containing 20 mM Tris–HCl (pH 7.5), 500 mM NaCl, and 4 mM $\beta$-mercaptoethanol, with the addition of 2 µg/ml RNase A and protease inhibitors. The resuspended cells were sonicated and centrifuged at high speed (48,000$g$) for 20 min; thus, the clear supernatant (cell lysate) was obtained. The cell lysate was bound with 3 ml of Ni-NTA beads (pre-equilibrated with 30 ml of T500). The beads were then washed with 30 ml of T500 plus 40 mM imidazole and added with 10 ml of T500 containing 200 mM imidazole to elute SRSF proteins. The eluted protein solution was added with T0 containing 20 mM Tris–HCl (pH 7.5), 0 mM NaCl, and 4 mM $\beta$-mercaptoethanol to get the final NaCl concentration of 100 mM. The diluted protein solution was loaded onto 0.5 ml of Q-Sepharose beads, which were subsequently washed with 30 ml of T150 containing 20 mM Tris–HCl (pH 7.5), 150 mM NaCl, and 2 mM DTT. The SRSF proteins were eluted from Q-Sepharose beads with 5 ml of T500 containing 20 mM Tris–HCl (pH 7.5), 500 mM NaCl, and 2 mM DTT. The purified proteins were quantified, examined in SDS–PAGE, and stored at −80°C until being used. The eluted SRSF7 protein from Q-Sepharose step was concentrated using the Centricon (30-kD cutoff) and then loaded on the gel filtration column (a Superdex 200 10/300 Gl column). The column was run using the Bio-Rad NGC Chromatography Systems in T500 with 10% glycerol added. The peaked fractions from gel filtration were collected, quality-checked by SDS–PAGE, quantity-measured by Bradford assays, and eventually stored at −80°C until being used.

To obtain the human MP (NLSD3-DGCR8) complex and DROSHA (D3-G2), we used the similar protocols described in our previous studies (Nguyen et al, 2015, 2020; Li et al, 2020). NLS was the nuclear localization sequence, containing seven amino acids (PKKKRKV). D3 was the DROSHA fragment, containing amino acids 390–1,365. G2 was the DGCR8 fragment containing amino acids 701–773. In short, pXab-NLSD3 and pXG-DGCR8 plasmids, or pXab-D3 and pXG-G2 plasmids, were cotransfected in the HEK293E cells. The NLSD3-DGCR8 or D3-G2 complex was purified by Ni-NTA beads and Q-Sepharose beads. The D3-G2 complex was further purified using NGC Chromatography Systems.

### Pre-miRNA synthesis

We synthesized pre-miR-142_CL-2 and CL0 as described in our previous study (Le et al, 2022). In short, we first generated the double-stranded DNA (dsDNA) containing the T7 promoter, the hammerhead (HH) ribozyme, a pre-miRNA sequence, and the hepatitis delta virus (HDV) ribozyme. Two oligos containing a part of HH ribozyme and a pre-miRNA sequence were annealed and then extended by Klenow Fragment (EP0421; Thermo Fisher Scientific) at 37°C for 2 h to make dsDNA. This dsDNA was amplified by the first PCR using two primers: one contained T7-HH ribozyme, whereas the other contained pre-miRNA and the 5′-half of HDV ribozyme sequence. Next, three other oligos were used to synthesize the dsDNA containing the HDV sequence, called HDV-dsDNA. This HDV-dsDNA and the dsDNA amplified in the first PCR were overlapped and amplified in the second PCR using the T7 primer and the HDV reverse primer to generate in vitro transcription (IVT)-dsDNA. Phusion Hot Start II DNA Polymerase (F549L; Thermo Fisher Scientific) was used for all PCRs, and the oligos were obtained from GENEWIZ and BGI Group. The PCR primer and template information are presented in Table S2.

Next, the IVTs were conducted with 200 ng of each dsDNA template using MEGAscript T7 Transcription Kit (AM1334; Invitrogen). The self-cleavage activity of the HH and HDV ribozymes released the pre-miRNAs, which contained 5′-OH and 2′-3′-cyclic phosphate. After being gel-purified, the 2′-3′-cyclic phosphate and 5′-OH were converted into 3′-OH and 5′-phosphate, respectively, by T4 polynucleotide kinase (EK0032; Thermo Fisher Scientific). Finally, the pre-miRNAs were purified using isopropanol and stored at −80°C until further use.

### In vitro pri-miRNA cleavage assays for individual human pri-miRNAs

We synthesized the dsDNA containing the T7 promoter and DNA region coding for each pri-miRNA by PCR. The PCR primer and template information are presented in Table S2. The sequences of these synthesized dsDNAs were confirmed by Sanger sequencing. Next, ~200 ng of each dsDNA was added in the IVT reactions containing the MEGAscript T7 Transcription Kit (AM1334; Invitrogen).

The resulting RNAs were gel-purified, quantified, and finally stored at −80°C.

To conduct the pri-miRNA cleavage assays, 2 pmol of each pri-miRNA were incubated with the enzymes at 37°C in 10 μl reaction buffer containing 50 mM Tris–HCl (pH 7.5), 150 mM NaCl, 2 mM $MgCl_2$, 1 mM DTT, 0.2 μg/μl BSA, and 10% glycerol. The amounts of enzyme were indicated in the figure and figure legends for each experiment. After 2 h, we treated the mixture with 20 μg of Proteinase K (Cat: 25530049; Invitrogen) for 15 min at 37°C and then 15 min at 50°C to stop the enzyme reaction. Finally, the mixtures were heated at 95°C for 5 min and immediately chilled on ice before being analyzed by a pre-run 12% urea-PAGE. The gel was run at 300 V for 90 min and was stained by SYBR Green II RNA Gel Stain (S7564; Invitrogen) for 5 min. The gel images were captured using the Bio-Rad Gel Doc XR+ system, and RNA band intensities were estimated using Image Lab v6.0.1. The relative pri-miRNA cleavage efficiency of MP or DROSHA was calculated as a ratio of the band density of cleaved products (F2) to that of the pri-miRNA substrate.

### The electrophoretic mobility shift assay (EMSA)

2 pmol of pri-miR-142, 20 pmol DROSHA, and 26 pmol of SRSF7 were mixed in 12 μl of EMSA reaction buffer containing 50 mM Tris-HCl (pH 7.5), 125 mM NaCl, 10% glycerol, 0.2 mg/ml BSA, 1 mM DTT, and 2 mM EDTA. Next, the EMSA reactions were incubated on ice for 1 h, and then, 5 μl of each reaction mixture was loaded onto a 4% native PAGE. The gel was run in the electrophoresis apparatus for 1 h 20 min at 4°C in a pre-cooled 1× TBE buffer. After completion of electrophoresis, the gel was stained with ethidium bromide for 5 min. The images of stained gels were taken using the Bio-Rad Gel Doc XR + system.

### The high-throughput pri-miRNA cleavage assay (HT-priCA)

We synthesized randomized pri-miR-142 and pri-miR-191 by the IVT. Each pri-miRNA possessed three randomized groups, containing four randomized base pairs in its CNNC position (group 1), and 4 nt upstream (group 2) or downstream (group 3) of its CNNC positions. Six groups of two randomized pri-miRNAs were synthesized similarly as described in the following for group 1 of pri-miR-142. First, two single-stranded DNAs, F-142lib (AAGAGTACACGCCGACGGACAGACA-GACAGTGCAGTC) and R-142lib_1 (GGCAGCAGTGGCGTGATCTCCNNNNCC-CACAGTACACTCATCCATAAAG), were used to amplify the dsDNA template of pri-miR-142 that was used in our previous study (Li et al, 2020) using Phusion Hot Start II DNA Polymerase (F549L; Thermo Fisher Scientific). The randomized nt were added in the R-142lib_1 primer. Next, the resulting dsDNA was amplified in the second PCR with F-T7-142lib primer (TAATACGACTCACTATA GGGAAGAGTACACGCCGACGGACAG) and R-142-RA3-1 (TTGGCACCCGAGAATTCCAGGCAGCAGTGGCGTGATCTCC). As a result, the two-round PCR-amplified dsDNA contained T7 promoter, pri-miR-142-group1–coding DNA, and RA3 sequence. These dsDNAs were subjected to the IVT reactions to produce the randomized pri-miRNAs. The IVT-synthesized RNAs were gel-purified, quality-checked by gel staining, quantity-checked by Nanodrop, and finally stored at −80°C until use. The PCR primer and template information are presented in Table S3.

In HT-priCA, 3 pmol of each randomized pri-miRNA group were incubated with 5 pmol of the MP with or without 10 pmol of SRSF3 or SRSF7 at 37°C for 2 h, in 10 μl of the standard reaction buffer as mentioned above. After the reaction, the cleaved RNA products were purified using phenol extraction followed by isopropanol precipitation. The purified RNAs were ligated with the RA5-6N adapter (GUU CAG AGU UCU ACA GUC CGA CGA UCN NNN NN), and then, the resulting RA5-6N–ligated RNAs were reverse-transcribed using R-RA3 primer and SuperScript IV Reverse Transcriptase (18090010; Invitrogen) to generate cDNAs. The cDNAs were finally amplified using PCR with the sequencing primers into DNA libraries of the cleaved products. The pri-miRNA substrates were cloned using a circularization method. The pri-miRNAs were first reverse-transcribed using SuperScript IV Reverse Transcriptase and cirRTP primer (5'Phos/NNN NNN GAT CGT CGG ACT GTA GAA CTC TGA AC/iSp18/CCT TGG CAC CCG AGA ATT CCA). The resulting cDNAs were circularized by CircLigase single-stranded DNA ligase (CL4115K; Lucigen). The circularized cDNAs were eventually amplified by PCR using the sequencing primers into DNA libraries of the pri-miRNA substrates. The DNA libraries were sequenced by NovaSeq 6000 sequencer, and the sequencing data were deposited in Gene Expression Omnibus (accession ID: GSE215790).

### High-throughput pri-miRNA cleavage library analysis

We removed adapters using cutadapt (-a TGGAATTCTCGGGTGCCAAGG-A GATCGTCGGACTGTAGAACTCTGAAC-m 10) (Martin, 2011). Then, we joined two paired-end reads using fastq-join (Aronesty, 2013). The low-quality and duplicated reads were filtered out using fastq_quality_filter (-q 20 -p 90) and fastx_collapser, respectively (http://hannonlab.cshl.edu/fastx_toolkit/index.html). The 6 and 4-nt randomized barcodes were discarded from the cleaved products and OS, respectively. We mapped the resulting reads to the built-in sequence references containing all 1,532 pri-miRNA variants using BWA (Li & Durbin, 2010). Only unique mapped reads were collected. The raw read counts were normalized as count per million.

The basal canonical cleavage sites of MP in pri-miR-142 and pri-miR-191 were named as CL according to the 3'-ends of 3p miRNAs (miRBase.org, Kozomara et al, 2019). The cleavage sites upstream and downstream of CL0 were labeled as CL-1 and CL1, respectively. The cleavage sites of MP in HT-priCA were determined by the 5'-ends of the cleaved products, F3 fragments. The cleavage site of MP was CLx when the 5'-end of F3 fragments was mapped to the position x-1 in pri-miRNA. The basal cleavage sites ranged from CL-3 to CL3. The apical cleavage sites were in between CL8 and CL14.

For each variant, we obtained the normalized counts of the original substrate ($N_S$), all cleaved products ($N_P$) derived from this substrate, and the normalized counts of a cleaved product ($N_X$) cleaved at a cleavage site, CLx, in which x is ranging from −3 to 3.

The global cleavage efficiency score of a variant ($E_P$) was the cleavage efficiency of the MP estimated for all the cleaved products resulting from this variant and was calculated using the following equation: $E_P = log_2(N_P + 0.1) − log_2(N_S + 0.1)$. A pseudocount of 0.1 was added.

The local cleavage efficiency score of a variant at the cleavage site of CLx ($E_X$) was estimated for only the cleaved product resulting from CLx cleavage and was calculated using the following equation: $E_X = log_2(N_X + 0.1) − log_2(N_S + 0.1)$. A pseudocount of 0.1 was added.

The cleavage accuracy score of the cleavage site CLx ($A_X$) was calculated for each variant as follows: $A_X = N_X/N_P$.

The $E_P$, $E_X$, and $A_X$ values were averaged using three scores from the three repeats for each cleaved product sample.

The relative cleavage accuracy and global and local cleavage efficiency scores for the MP + cofactor sample were calculated for each variant as follows:

$$\Delta A_{X\,[cofactor]} = A_{X\,[MP + cofactor]} - A_{X\,[MP]},$$

$$\Delta E_{P\,[cofactor]} = E_{P\,[MP + cofactor]} - E_{P\,[MP]},$$

$$\Delta E_{X\,[cofactor]} = E_{X\,[MP + cofactor]} - E_{X\,[MP]}.$$

The structures of pri-miRNA variants were predicted using RNAfold (Lorenz et al, 2011).

### Human pri-miRNA cleavage library analysis

We downloaded the cleavage efficiency and homogeneity scores of 1,881 pri-miRNAs cleaved by MP alone or MP and SRSF3 (Kim et al, 2021). In the original study, cleavage efficiency scores were calculated as the ratio of the productive cleaved products (±3 nt from the miRBase annotations of 5'-end of 5p miRNAs or 3'-end of 3p miRNAs) to the OS. The cleavage homogeneity scores were calculated as the maximum cleavage ratio of all productive sites. We selected 788 pri-miRNAs cleaved by MP and SRSF3 with cleavage efficiency and homogeneity scores higher than 1 and 0.1, respectively. The CNNC and 17-CRC motifs were detected from positions −15 to −22 and positions −17 to −19 on the 3'-strand of these 788 pri-miRNAs, respectively.

### Human pri-miRNA structure analysis

We extracted 504 pri-miRNA sequences from the human genome reference (hg38; https://genome.ucsc.edu/) using the genome annotation from MirGeneDB v2.0 (Fromm et al, 2020). Each pri-miRNA sequence includes its pre-miRNA and 40-nt extensions at both ends of pre-miRNA. The structure of each pri-miRNA was predicted using RNAfold (Lorenz et al, 2011). We found 332/504 pri-miRNAs containing the CNNC motif from positions −15 to −22 on their 3'-strand. The CNNC structures of these 332 pri-miRNAs were examined.

### SHAPE-MaP library construction

RNAs were incubated in a buffer containing 50 mM Tris–HCl (pH 7.5), 100 mM NaCl, and 2 mM MgCl$_2$ at 37°C for 20 min. Next, 1M7 (1-methyl-7-nitroisatoic anhydride) (908401; Sigma-Aldrich) was added to the concentration of 10 mM and incubated for 75 s at 37°C to modify RNAs. The control samples were prepared in parallel, where DMSO was added instead of 1M7. After modification, RNAs were precipitated by isopropanol and washed with ethanol before being ligated to the 3' adapter (6N-RA3). Next, the reverse transcription was performed for 3 h at 42°C by SuperScript II (18064014; Invitrogen) using R-RA3 primer in the buffer containing 0.5 mM dNTPs, 50 mM Tris–HCl (pH 8.0), 75 mM KCl, 6 mM MnCl$_2$, and 10 mM DTT. The resulting cDNAs were later amplified two times, the first time by R-RA3 and the pri-miR-142 sequence-specific

forward primers, and the second time by two sequencing primers, using Phusion Hot Start II DNA Polymerase. The final DNA products were subjected to NGS using the Illumina NovaSeq 6000 sequencer. The primer information for pri-miRNA cloning in SHAPE-MaP experiments is presented in Table S4. The sequencing data of SHAPE-MaP experiments were deposited in Gene Expression Omnibus (accession ID: GSE221996).

### SHAPE-MaP analysis

We removed adapters, joined two paired-end reads, and removed low-quality and duplicated reads as described for the HT-priCA. Then, the 6-nt randomized barcodes in the 3'-end of the remaining reads were discarded. Next, ShapeMapper2 software was used to identify RNA structures of pri-miR-142 variants (shapemapper –min-seq-depth 1,000) (Busan & Weeks, 2018). The resulting SHAPE reactivity data from ShapeMapper2 were applied to detect RNA structures using the RNAstructure package (Reuter & Mathews, 2010).

### iLIME-seq for pre-miRNA

1 μg of pcDNA3-CNNC-pri-miR-142 or pcDNA3-non-CNNC-pri-miR-142 plasmid were transfected into one well in a six-well plate of HCT16-DICER KO cells by Lipofectamine 3000 Transfection Reagent (L3000001; Invitrogen). The primer information for cloning pcDNA3-pri-miR-142 plasmids is presented in Table S1. The cells were harvested after 2 d. In another experiment, 1.5 μg of pXG, pXG-SRSF7, or pXG-SRSF3 were cotransfected with 1 μg of pcDNA3-CNNC-pri-miR-142 into one well in the six-well plate of HCT116-DICER KO cells. The cells were also harvested 2 d later. All the total RNAs were extracted using TRIzol Reagent (15596018; Invitrogen). We used iLIME-seq to clone and sequence pre-miR-142 from these total RNAs (Le et al, 2022). In short, 2.5 μg of total RNA were incubated with 16 units of T4 RNA ligase 1 (M0437M; NEB) in the ligation buffer containing 50 mM Tris–HCl (pH 7.5), 10 mM MgCl$_2$, 1 mM DTT, 16 units of SUPERase·In RNase inhibitor (AM2694; Invitrogen), and 20% PEG 8000. The mixtures were incubated at 25°C for 2 h. The ligated RNA products were then subjected to reverse transcription reactions using SuperScript IV (Reverse Transcriptase) at 50°C for 20 min. The resulting cDNAs were amplified the first time by pre-miR-142 sequence-specific primers and the second time by sequencing primers, both by Phusion Hot Start II DNA Polymerase (F549L; Thermo Fisher Scientific). The final DNA products from the second PCR were subjected to NGS using the Illumina NovaSeq 6000. The sequencing data were deposited in Gene Expression Omnibus (accession ID: GSE215791). The primer information for pre-miR-142 iLIME experiments is presented in Table S5.

We processed sequencing reads and mapped reads to build-in reference for iLIME-seq samples as described in the previous study (Le et al, 2022).

## Data Availability

The sequencing data from this publication were deposited to the Gene Expression Omnibus database (HT-pri-miRNA cleavage experiments, accession ID: GSE215790; SHAPE-MaP experiments, accession ID: GSE221996; and iLIME-seq experiments, accession ID: GSE215791).

# Supplementary Information

# Acknowledgements

This work was supported by the Croucher Foundation [CIA17SC03]. TD Nguyen and MN Le are recipients of the Hong Kong PhD Fellowship Scheme. We are grateful to Dr. Narry V Kim (Seoul National University, Korea) for sharing with us the HCT116 DICER-KO cells. We also appreciate our laboratory members, especially Chen Hui, for their discussion and technical assistance.

## Author Contributions

MN Le: data curation, formal analysis, validation, visualization, methodology, and writing—original draft, review, and editing.
TD Nguyen: data curation, software, formal analysis, validation, visualization, methodology, and writing—original draft, review, and editing.
TA Nguyen: conceptualization, data curation, supervision, funding acquisition, investigation, project administration, and writing—original draft, review, and editing.

## Conflict of Interest Statement

The authors declare that they have no conflict of interest.

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
