## [Reviewer comments · Life Science Alliance]

SRSF7 and SRSF3 depend on RNA sequencing motifs and secondary structures to regulate Microprocessor

Minh Ngoc Le, Trung Duc Nguyen and Tuan Anh Nguyen

DOI: 10.26508/lsa.202201779

Corresponding author(s): Prof. Tuan Anh Nguyen (Hong Kong University of Science and Technology)

Review timeline:

Submission Date:	2022-10-20
Editorial Decision:	2022-11-22
Revision Received:	2023-01-05
Editorial Decision:	2023-01-23
Revision Received:	2023-01-25
Accepted:	2023-01-26

Scientific Editor: Eric Sawey

Transaction Report:

No Peer Review Process File is available with this article, as the authors have chosen not to make the review process public in this case.

22 November 2022

1st Editorial Decision

Re: Life Science Alliance manuscript #LSA-2022-01779-T

Prof. Tuan Anh Nguyen
Hong Kong University of Science and Technology
Clear Water Bay, Kowloon
Hong Kong
China

Dear Dr. Nguyen,

Thank you for submitting your manuscript entitled "SRSF7 and SRSF3 depend on RNA sequencing motifs and secondary structures to regulate Microprocessor" to Life Science Alliance. The manuscript was assessed by expert reviewers, whose comments are appended to this letter. We invite you to submit a revised manuscript addressing the Reviewer comments.

Thank you for this interesting contribution to Life Science Alliance. We are looking forward to receiving your revised manuscript.

Sincerely,

B. MANUSCRIPT ORGANIZATION AND FORMATTING:

RE: Life Science Alliance Manuscript #LSA-2022-01779-TR

Prof. Tuan Anh Nguyen
Hong Kong University of Science and Technology
Clear Water Bay, Kowloon
Hong Kong, China

Dear Dr. Nguyen,

Thank you for submitting your revised manuscript entitled "SRSF7 and SRSF3 depend on RNA sequencing motifs and secondary structures to regulate Microprocessor". We would be happy to publish your paper in Life Science Alliance pending final revisions necessary to meet our formatting guidelines.

- please address Reviewer 3's remaining points
- we encourage you to introduce your panels in your figure legends in alphabetical order
- the GEO datasets should be made publicly accessible at this point. Please remove the Reviewer Access codes from your Data Availability Statement.

A. FINAL FILES:

-- Summary blurb (enter in submission system): A short text summarizing in a single sentence the study (max. 200 characters including spaces). This text is used in conjunction with the titles of papers, hence should be informative and complementary

to the title. It should describe the context and significance of the findings for a general readership; it should be written in the present tense and refer to the work in the third person. Author names should not be mentioned.

B. MANUSCRIPT ORGANIZATION AND FORMATTING:

Sincerely,

3rd Editorial Decision

26 January 2023

RE: Life Science Alliance Manuscript #LSA-2022-01779-TRR

Prof. Tuan Anh Nguyen
Hong Kong University of Science and Technology
HKUST, Clear Water Bay, Kowloon
Hong Kong, China

Dear Dr. Nguyen,

Thank you for submitting your Research Article entitled "SRSF7 and SRSF3 depend on RNA sequencing motifs and secondary structures to regulate Microprocessor". It is a pleasure to let you know that your manuscript is now accepted for publication in Life Science Alliance. Congratulations on this interesting work.

DISTRIBUTION OF MATERIALS:

Again, congratulations on a very nice paper. I hope you found the review process to be constructive and are pleased with how the manuscript was handled editorially. We look forward to future exciting submissions from your lab.

Sincerely,
